# Technologies for Viable Circulating Tumor Cell Isolation

**DOI:** 10.3390/ijms232415979

**Published:** 2022-12-15

**Authors:** Maria S. Tretyakova, Maxim E. Menyailo, Anastasia A. Schegoleva, Ustinia A. Bokova, Irina V. Larionova, Evgeny V. Denisov

**Affiliations:** 1Laboratory of Cancer Progression Biology, Cancer Research Institute, Tomsk National Research Medical Center, Russian Academy of Sciences, 634009 Tomsk, Russia; 2Single Cell Biology Laboratory, Research Institute of Molecular and Cellular Medicine, Peoples’ Friendship University of Russia (RUDN University), 117198 Moscow, Russia

**Keywords:** CTC, isolation, cancer, metastasis, in vitro, in vivo

## Abstract

The spread of tumor cells throughout the body by traveling through the bloodstream is a critical step in metastasis, which continues to be the main cause of cancer-related death. The detection and analysis of circulating tumor cells (CTCs) is important for understanding the biology of metastasis and the development of antimetastatic therapy. However, the isolation of CTCs is challenging due to their high heterogeneity and low representation in the bloodstream. Different isolation methods have been suggested, but most of them lead to CTC damage. However, viable CTCs are an effective source for developing preclinical models to perform drug screening and model the metastatic cascade. In this review, we summarize the available literature on methods for isolating viable CTCs based on different properties of cells. Particular attention is paid to the importance of in vitro and in vivo models obtained from CTCs. Finally, we emphasize the current limitations in CTC isolation and suggest potential solutions to overcome them.

## 1. Introduction

CTCs are cells that detach from the primary tumor site and enter the bloodstream [1]. According to the “seed and soil” theory, these cells are defined as “seeds” that can cause metastasis [1]. The identification of CTCs remains a technical challenge due to their extremely high phenotypic heterogeneity and the low representation of these cells in the bloodstream (1–10 cells per billions of other blood cells) [2]. Even after successful isolation, the maintenance of tumor cell viability remains a key issue. Thus, effective approaches with high sensitivity and specificity are urgently needed for the isolation of CTCs [3,4,5].

The isolation of viable CTCs is useful for investigating the mechanisms of metastasis and improving anticancer treatment [6]. The preservation of cell integrity and viability also provides information about the genetic and molecular features of CTCs. In particular, a sufficient amount of viable CTCs is needed for multiomic analysis, namely, single-cell sequencing, to obtain reliable data that reflect the characteristics of CTCs in the body. Obtaining viable CTCs enables the creation of preclinical models for drug screening and modeling of the metastatic cascade, including migration, invasion, and extravasation [7,8]. CTC-derived cell lines provide an opportunity to identify new markers for isolating CTCs from the blood and to develop additional predictive and prognostic criteria [9]. CTC cultures allow us to evaluate the response to anticancer therapeutics and to individualize the treatment as well as opening up doors for the development of drugs aimed at the prevention of metastasis. CTC cell lines and xenografts (CDX) are also necessary to better understand the functional properties of CTCs obtained from cancer patients and to identify metastasis-initiating cells [10]. In general, CTC in vitro, ex vivo and in vivo models provide opportunities to determine the biological properties of primary tumors and future metastatic cells, thus opening new horizons for basic and translational research.

The existing methods for isolating CTCs can be divided into two main types: isolation based on physical factors (centrifugation and filtration) and isolation based on immunoaffinity, i.e., the presence of specific proteins on the cell surface [11]. These methods are not mutually exclusive and can be combined [12]. The CellSearch system, which is based on immunoaffinity, was the first to be approved by the US Food and Drug Administration (FDA) in 2004, and it is the most commonly used method for capturing CTCs [13,14,15]. However, CellSearch isolates cells that only express common epithelial markers—EpCAM and cytokeratins (CK 8, 18, and/or 19). CTCs that are negative for non-epithelial markers, for example, due to epithelial–mesenchymal transition, are not detected by this method. Moreover, the frequency of capturing dead cells by CellSearch is high [16]. In contrast, other methods, such as different microfluidic systems and filter-based approaches, do not depend on biological markers and isolate a high percentage of viable CTCs [17,18,19]. However, the standardization and clinical implementation of the developed technologies for the isolation of CTCs remain important tasks in cancer research and clinical practice.

The present review provides an overview of the current state of the development of relevant methods for isolating CTCs with high throughput and viability. Methods that are not suitable for obtaining viable CTCs such as CellSearch, Ariol system, AdnaTest, and others are not reviewed here.

## 2. Circulating Tumor Cells: Features and Clinical Significance

The analysis of CTCs is an effective instrument for understanding the mechanisms of cancer metastasis [20]. However, only limited CTCs survive and invade distant sites [21]. The identification of such metastasis-associated CTCs is one of the key challenges and can reveal molecules that are critical for metastasis and for the development of antimetastatic therapy.

CTCs possess high heterogeneity, which is similar to primary tumor cells [22]. CTCs have specific physical, genetic, and phenotypic properties that differentiate them from blood cells (Figure 1) [21]. CTCs are larger (8–25 μm) than normal blood cells (5–20 μm) [23,24] and demonstrate higher unit membrane capacitance and lower cytoplasm conductivity as compared to leukocytes [25]. Similar to blood cells, CTCs possess high deformability, which is related to their ability to undergo epithelial–mesenchymal transition [26,27].

CTCs are also represented by hybrid cells that are formed by the fusion of tumor cells with other cells, such as leukocytes, macrophages, fibroblasts, and mesenchymal stem cells. Circulating hybrid cells demonstrate high viability in the bloodstream due to their ability to avoid immune recognition, increased drug resistance, and metastatic potential [28].

CTCs express surface adhesion molecules of epithelial origin, such as EpCAM and various cytokeratines (e.g., CK 5, 7, 8, 18, and 19) [22,29]. However, CTCs may also have a mesenchymal phenotype or possess epithelial/mesenchymal hybrid properties [30]. For instance, CTCs can simultaneously express both epithelial (EpCAM and/or CK) and mesenchymal markers (N-cadherin, Vimentin, Twist, Snail, Zeb, and others) [31]. In addition, CTCs can harbor “stemness” properties, whose identification is based on the assessment of the expression of CD44, CD24, ALDH1, and CD133 proteins [32,33,34]. CTCs can also have cancer-specific markers such as human epidermal growth factor receptor 2 (HER2), estrogen receptors, prostate-specific membrane antigen and others [20].

In the bloodstream, the majority of CTCs are influenced by detrimental shear stress or they undergo anoikis, a programmed cell death due to the detachment of the cell [20]. Several CTCs interact with platelets, neutrophils, macrophages, myeloid-derived suppressor cells, or cancer-associated fibroblasts to escape the immune system and enhance their survival [20,35,36].

Although detecting CTCs in peripheral blood is challenging, CTCs are highly significant in clinical applications. The presence of CTCs is a prognostic factor in many cancers [37,38,39,40,41]. The amount of CTCs correlates with cancer aggressiveness, increased metastasis risk, and frequency of relapses [42,43,44,45,46]. CTCs expressing mesenchymal or stemness-related markers are associated with poor survival [47,48]. CTCs are also potential biomarkers for monitoring the response to anti-cancer therapy [49,50,51]. Several studies have identified CTCs at the early stages of tumor development, which demonstrates their importance for early cancer diagnosis, and shown that CTCs provide clinically important information to map tumor heterogeneity and tumor evolution [52,53,54,55].

## 3. Methods for the Isolation of Viable CTCs

CTC isolation methods are based on the cell’s physical properties (e.g., size and surface charge) and biological features, i.e., expression of specific surface proteins [56]. Methods based on the combination of these properties have also been developed (Figure 2, Table 1, Table 2 and Table 3).

### 3.1. Isolation of CTCs Based on Cell Physical Properties

Methods based on the physical factors (Figure 2, Table 1) allow the isolation of viable CTCs with high capture efficiency without using fluorescent labels [57]. These so-called “label-free” methods attract attention because no cell loss is observed as compared to methods that use CTC-specific markers [58]. The isolation of CTCs based on physical properties helps to distinguish CTCs from other cells in peripheral blood by cell size and electrical properties [57]. The main obstacle of these methods is associated with the clogging of mechanical microfilters and microfluidic systems and with the adhesion of peripheral blood cells to the filter surface [57]. The increased fluid pressure inside the filters can damage the cells. In addition, long-term contact with the filter surface can lead to irreversible adhesion of trapped cells, resulting in reduced cell isolation efficiency [59].

Different size-based methods for viable CTC isolation have been developed, most of which are based on the use of membrane microfilters and microfluidic technologies. The isolation by size of epithelial tumor cells (ISET) system with a modified filtration buffer isolates viable CTCs with no antibody-related bias and no/minimum cell loss. This method detects both EpCAM^+^ and EpCAM^−^ CTCs. ISET is very labor intensive and can also enrich fixed CTCs [60,61]. Viable CTCs can also be isolated by two other devices, one of which consists of a three-dimensional (3D) palladium (Pd) filter with 8 µm-sized pores in the lower layer and a 30 µm-sized pocket in the upper layer, whereas the other device (MetaCell) is based on the pass of peripheral blood through a porous polycarbonate membrane [62,63]. The 3D Pd filter can isolate EpCAM^+^CD45^−^ CTCs, which present as small clusters and single cells, as well as cells with EpCAM^+^CD45^+^ and EpCAM^−^CD45^−^ phenotypes. However, this filter is difficult to manufacture and requires high-precision lithography with electroforming technology [62]. MetaCell and 3D Pd filter ensure the high viability and active growth of CTCs in vitro [62,63,64,65]. Another size selection method is the centrifugation-force-based CD-PRIME platform that captures CTCs selectivity by their size on a membrane in a chamber. After membrane removal, these CTCs can be used for in vitro experiments [66]. CTCs isolated by this method include cells with EpCAM^+^CK^+^CD45^−^ and EpCAM^−^CK^−^CD45^+/−^ phenotypes. This platform is fully automatic and easy to use, but requires special equipment [66]. The effective capture of viable CTCs was also demonstrated by fluid-assisted separation technology (FAST), which selectively separates CTCs by size through the pores in the membrane [19,67], and the ScreenCell method based on non-invasive blood filters for CTC and CTC cluster enrichment [68]. The ScreenCell device contains 7.5 μm pores and uses a vacutainer to transfuse blood through the filter from the upper chamber [68]. The composition of the cells after isolation by the ScreenCell included cells with the pan-CK^+^/CK7^+^CD45^−^ phenotype [68]. This method does not require expensive equipment and special operator skills. Another 3D microfilter device consists of two layers of parylene-C (poly(monochloro-p-xylylene) membrane with pores and ensures the capture of viable CTCs [69].

CTCs can be isolated by microfluidic chips with micro-ellipse filters [70]. This approach is based on the transport of blood samples through the ellipse constriction matrix, preventing the entry of large, rigid tumor cells and allowing small plastic cells to pass. The application of frictionless gradual micro-ellipse filters provides highly reproducible and sensitive capture of CTCs and ensures their viability [70]. The Parsortix is an epitope-independent microfluidic system composing of a microscale stepped spacer structure with a cross-sectional gap that progressively reduces the size of the fluid path [71,72,73]. The Parsortix allows the isolation of viable CK^+^CD45^−^CTCs in a liquid suspension with extremely high purity for further molecular and functional analysis [17,74]. However, cells can be deformed during isolation under the action of mechanical forces. The CTCs isolated with the Parsortix are large (15.6 ± 2.0 μm), suggesting that the Parsortix may miss small CTCs [75]. Size-based spiral microfluidic technology can also isolate the viable CTCs through the use of hydrodynamic forces that present in the curvilinear microchannels [76]. This technology is capable of the high-speed processing of large volumes of blood and isolating CTC clusters with elevated viability [77,78]. The deterministic lateral displacement (DLD) method utilizes a periodically-arranged micropillar array to produce a specific streamline pattern. Cells that are larger than the critical point can be displaced from their original lateral position at the device inlet. Cells that are smaller than the critical point move in a zigzag mode through the pillar array [79]. This method provides high-throughput and clog-free isolation through a cascaded microfluidic design [80]. The present platform not only isolates CTCs, but also identifies tumor fusion (hybrid) cells and enables the analysis of CTCs at single-cell resolution [80]. The challenge of CTCs of different sizes has been overcome by a microfluidic device, the “Labyrinth”, which utilizes inertial forces to focus CTCs into separate streamlines. This differential focus is formed by the balance between the inertial lift and Dean forces, which affects the cells so that they flow in the direction of the focusing stream [81,82]. The single-cell immunoblotting (ieSCI) microfluidic system based on zigzag channel-based label-free and high-efficiency cell sorting was shown to isolate CTCs from breast cancer patients. These CTCs were used for direct analysis or single-cell immunoblotting [83]. Other studies showed that an inertial microfluidic chip can isolate viable CTCs from the blood and seminal fluid of prostate cancer patients [84,85].

**Table 1 ijms-23-15979-t001:** Cell size-based methods for isolating viable CTCs.

Methods		Cancer Type	Further Applications	Advantages (+)/ Disadvantages (−)	References
Membrane microfilters	ISET	Non-small cell lung, colorectal cancers, melanoma	Whole-genome sequencing	+ 80–90% sensitivity + Isolation of CTC clusters − High loss of small cells	[60,61,86,87]
	3D palladium filter	Breast cancer	FISH, KRAS mutation analysis	+ 85% cell purity – High loss of small cells	[62]
	MetaCell	Colorectal, lung cancers	CTC culturing, gene expression analysis	+ Depletion of more than 95% of leukocytes + Isolation of CTC clusters – High loss of small cells	[18,65,88]
	CD-PRIME	Pancreatic cancer	NA	+ 76% sensitivity – Contamination with leukocytes – High loss of small cells	[66]
	FAST	Colorectal, breast, stomach, lung cancers	KRAS mutation analysis	+ 95.9 ± 3.1% sensitivity – High loss of small cells	[19,67]
	ScreenCell	Laryngeal, pancreatic cancers	FISH, ddPCR	+ Fast isolation (<30 min) + Isolation of CTC clusters – High loss of small cells	[89,90]
	3D microfilter device	NA	NA	+ 86.5 ± 5.3% capture efficiency – High loss of small cells	[69]
Microfluidic technologies	Micro-ellipse filter	Breast, colorectal, non-small cell lung cancers	Immunofluorescence analysis	+ 90% capture efficiency – Contamination with leukocytes – Clogging of filters	[70,91]
	Parsortix	Breastcancer	Mouse xenograft models, transcriptome or genome analysis, FISH	+ 66–96% capture efficiency + Isolating CTC clusters – Contamination with blood cells – High loss of small cells	[71,72,73]
	Spiral microfluidic technology	Glioblastoma	FISH	+ Fast isolation (15 min) + Isolation of CTC clusters + 90% recovery rate + 1.7 mL min processing rate – High loss of small cells	[76,77]
	DLD	Lung cancer	Transcriptome analysis	+ 96% capture efficiency + Depletion of more than 99% of leukocytes + 98% viability + 1 mL min processing rate – High loss of small cells – Contamination with blood cells	[79,80]
	Labyrinth	Non-small cell lung, liver cancers	FISH	+ Isolation of CTC clusters + Depletion of more than 95% of leukocytes – High loss of small cells	[81,92]
	ieSCI-chip	Breastcancer	Protein analysis at single-cell resolution	+ 89.92 ± 3.37% capture efficiency + Fast isolation (6 min) + 1.4 mL min processing rate	[83]
	Inertial microfluidic device	Prostate, laryngeal, thyroid, floor of the mouth, non-small cell lungcancers	NA	+ 89%±3.8% capture efficiency + Fast isolation (20 min) – Contamination with blood cells	[84,85]
Dielectric permittivity	DEP-FFF	NA	Immunofluorescence, FISH, and gene mutation analysis	+ 70–75% capture efficiency – High loss of small cells	[93,94]
	ODEP	Head and neck cancer	NA	+ 81.6–86.1% cell purity + Isolation of CTC clusters – High loss of small cells	[95,96]

NA, not available.

In contrast to the various size-based methods, there are other approaches for CTC isolation that focus on cell surface charge. For example, the dielectrophoretic field-flow-fractionation (DEP-FFF) method isolates viable CTCs at a high rate due to their dielectric properties, which are different from those of non-tumor cells [93]. A similar approach for viable CTC capture is negative selection followed by optically-induced dielectrophoresis in a microfluidic chip (ODEP). This device significantly increases the purity of the resulting fraction of CTCs while maintaining their viability. The ODEP method isolates EpCAM^+^ CTCs and EMT-transformed cancer cells [95].

Thus, size-based methods are the most common approach for CTC isolation. These methods have a high throughput but the main limitation is the heterogeneity of the CTCs size. One promising method is the capture of CTCs based on their dielectric properties, providing a very good bandwidth and purity. However, the main disadvantage is the presence of Joule heating, which occurs when fluid flows through pores and can lead to cell damage and decrease the sensitivity of the device [97].

### 3.2. Isolation of CTCs Based on Cell Biological Properties

Methods based on cell biological properties (Figure 2, Table 2) include positive selection with targeting markers on tumor cells or negative selection and targeting common non-tumor markers to remove other cells [56]. Techniques based on positive selection are less effective for the isolation of CTCs with low EpCAM and CK expression, which have been described in several cancers [98,99].

Besides the CellSearch system mentioned above, there are other marker-based methods for the positive selection of CTCs. For example, CTCs can be captured by magnetic nanoparticles coated with an antifouling hydrogel layer that inhibits the adhesion of nonspecific cells. EpCAM antibodies covalently grafted onto the surface of the hydrogel layer provide high specificity for CTC capture (MNPs@hydrogel-anti-EpCAM). Testing this method on mimic clinical blood demonstrates high specificity, velocity capture, and CTC viability after glutathione treatment [100]. An aptamer-trigger-clamped hybridization chain reaction (atcHCR) method has been developed for in situ identification and the subsequent cloaking/decloaking of CTCs by porous DNA hydrogels. This method captures viable CTCs directly through the EpCAM with minimal cell damage [101]. The release of cells from the hydrogel is quite gentle, thus the isolated cells are not damaged.

Currently, there are no universal markers that can be used to detect and enrich all CTCs. For example, CTCs undergoing EMT are EpCAM-negative and cannot be isolated by an anti-EpCAM specific assay [21]. In this case, negative selection, e.g., CD45 depletion, is a more optimal approach for CTC isolation [56]. A well-known approach for negative selection of CTCs is the RosetteSep, which obtains viable single CTCs and CTC clusters by removing unwanted CD45-positive cells bound to tetrameric antibodies and precipitated in a Ficoll-Paque density gradient by centrifugation [68]. However, cell isolation by the RosetteSep takes a long time and requires manual processing, thus limiting the robustness and reproducibility of the results. In addition, a new approach for the isolation of mesenchymal CTCs based on magnetic nanoparticles has been recently proposed [102]. In particular, N-cadherin-conjugated magnetic nanoparticles (NP@MNP) via biotin and streptavidin interactions have high capturing efficiency and can maintain cell viability [102].

The undoubted advantage of positive selection is the high purity of the resulting CTC fraction, and the disadvantage is the impossibility of using a panel of markers that can cover all CTC populations. On the contrary, the advantage of methods based on negative selection is the capture of all possible populations of CTCs, but at the same time, a large number of non-target cells are isolated.

### 3.3. Combined Methods

CTC isolation may also be based on both the cell’s physical and biological features [103]. Such combined methods can be based on microfluidics systems or without them (Figure 2, Table 3). A microfluidic RBC (red blood cells) chip has been developed based on the DLD model. In this device, CTCs are captured by a layer of RBCs with modified EpCAM antibodies on their surface and with dual-terminal functionalized DNA, 5′-Chol-DNA-Biotin-3′ as linkers [104]. Altered erythrocytes have a high affinity that enables them to recognize and capture CTCs effectively.

Moreover, the layer of RBCs prevents CTCs from damage during cell-microcolumn collision. CTCs can be gently released by destruction of the RBC layer with lysis buffer [104].

**Table 2 ijms-23-15979-t002:** Methods for isolating viable CTCs based on cell biological properties.

Methods		Cancer Type	Further Applications	Advantages (+)/Disadvantages (–)	References
Positive selection	MNPs@hydrogel-anti-EpCAM	NA	NA	+ 96 % viability + Fast isolation (25 min) – Isolation of only EpCAM-positive CTCs	[100]
	atcHCR	NA	NA	+ Isolating CTC clusters – Isolation of only EpCAM-positive CTCs	[101]
	NP@MNPs	NA	scRNAseq	+ Capture of mesenchymal CTCs + 85% capture efficiency – Isolation of only N-cadherin-positive CTCs	[102]
Negative selection	RosetteSep	Liver, breast cancers	scRNAseq, CTC culturing	+ Isolation of CTC clusters + Fast isolation (40 min) + CTC marker-free isolation – High number of untargeted cells	[68,105,106]

NA, not available; scRNAseq, single-cell RNA sequencing.

**Table 3 ijms-23-15979-t003:** Combined methods for isolating viable CTCs.

Methods		Cancer Type	Further Applications	Advantages (+)/Disadvantages (–)	References
Based on cell biological properties and microfluidic approaches	RBC-chip	Colorectal cancer	ddPCR	+ 96.5% sensitivity and specificity + 96.1% viability – Isolation of only EpCAM-positive CTCs	[104]
	CTC-iChip	Breast, prostate, lung cancers, melanoma, glioma	CTC culturing	+ 89.2 ± 5.7% capture efficiency + Capable of enrichment of CTCs with either positive or negative selection – Multistep nature of the protocol – Loss of CTCs associated with leukocytes	[107,108,109]
	HER2-GEDI	Breast, gastric cancers	CTC culturing	+ No any blood processing is required + Utilization of only 1 mL of blood – Isolation of only HER2-positive CTCs	[110]
	Isoflux	Gastroesophageal cancer	KRAS mutation analysis	+ 87% capture efficiency – Isolation of only EpCAM-positive CTCs	[111,112]
	Herringbone-Chip	Lung cancer	CTC culturing	+ One step method + 95% capture efficiency + Isolation of CTC clusters – Isolation of only EpCAM- and EGFR-positive CTCs	[113]
	Fluorescent microspheres	Colorectal, breast, non-small cell lung cancers	NA	+ 90% capture efficiency	[91]
Based on cell biological properties and non-microfluidic approaches	AccuCyte- RareCyte/PIC & RUN	Prostate, breast, lung cancers	scRNAseq, CTC culturing	+ 90% capture efficiency + 91.6% viability – The presence of false-positive CTCs	[114,115]
	Neu-IMNs	Breast cancer	PCR, Sanger sequencing, CTC culturing	+ 96.82% capture efficiency + 90.68% purity – Contamination with leukocytes	[116]

NA, not available; scRNAseq, single-cell RNA sequencing.

The method isolates EpCAM^+^CD45^−^CTCs with a high capture efficiency. Another method, the CTC-iChip system, is an inertial separation device for the removal of RBCs and platelets followed by high-gradient magnetic cell sorting for the depletion of white blood cells [107,109]. This system is fully automatic and is not subject to operator error, but it requires a long time for sample processing (6–7 h) [107]. Combined methods for the isolation and detection of viable CTCs also include microfluidic methods with immunomagnetic microbeads coated with CTC-specific markers, for example, the Herringbone-Chip and Isoflux systems with anti-EpCAM and EGFR microbeads and a HER2-based microfluidic device [110,117]. Furthermore, optical force and fluorescent microspheres have been integrated to isolate the viable CTCs. Fluorescent microspheres with a high-refractive index, capture and transfer CTCs to the collection channel, where the latter are detected by a fluorescent microscope and sorted [118].

CTCs can also be isolated by centrifugation in AccuCyte separation tubes followed by CTC labeling (EpCAM, HER2 and EGFR) and detection on glass slides using fluorescence and isolation by needle (RareCyte) [115]. This combined method, which is called PIC&RUN, demonstrates good performance due to its use of the positive and negative selection of CTCs. However, the main limitation of this method is the relatively long processing time (more than 2 h) due to semi-automatic scanning [115]. Another promising strategy is to coat neutrophil membrane-derived vesicles onto the high-performance biomimetic immunomagnetic nanoparticles (Neu-IMNs), which increases the efficiency of CTC isolation by enhancing the interaction between CTCs and neutrophils and improving their viability through soft interaction between neutrophil membranes [116]. This method isolates CK^+^CD45^−^CTCs, is easy and inexpensive, and only takes about an hour.

Thus, the combined approaches are partially capable of solving the problems of methods based on cell physical and biological properties, but they are not without drawbacks and still need to be optimized to improve specificity, the sensitivity, and throughput of CTC isolation.

## 4. Viable CTCs in In Vitro, Ex Vivo and In Vivo Studies

CTC-derived in vitro models are an effective instrument for the development of metastasis-targeting therapy [88,119,120]. Several studies have demonstrated successful examples of the establishment of cell cultures from CTCs [88,113]. For instance, gastrointestinal CTC lines (UWG01CTC and UWG02CTC) have been obtained using a microfluidic device IsoFlux, and completely reflect the genotypic and phenotypic heterogeneity of CTCs [121]. CTC-iChip-derived breast cancer lines were maintained in vitro for >6 months. Cultured CTCs shared cytological features with the matched primary CTCs, but had an increased proliferative signature [119].

Lung CTC line CTC-TJH-01, which was obtained by combined microfluidic Herringbone-Chip and immunomagnetic microbeads, demonstrated an intermediate epithelial/mesenchymal phenotype and the stem cell-like characteristics of CTCs. The analysis of this cell line showed a high expression of CXCL5 and low expression of CX3CL1 as a mechanism of CTC escape from the immune system [113]. Successful CTC cultures have also been developed from esophageal, bladder and pancreatic cancers using the MetaCell method [18,64,122]. Nine permanent lines of colon cancer were obtained from CTCs collected at different points in the treatment time using the RosetteSep Human Circulating Epithelial Tumor Cell Enrichment Cocktail. These cell lines showed different gene expression profiles and reflected changes in the cell phenotypes during therapy [123,124,125].

CTCs are also used to develop ex vivo models that are cultivated on biological tissue in an artificial environment with minimal change in natural conditions. However, such models are not without limitations. In particular, an ex vivo model with nanoemulsions to preserve the natural conditions, obtained from RosetteSep-derived breast cancer CTCs lost epithelial features and acquired stem or mesenchymal properties, as well as showed a gene expression profile that was distinct from the primary CTCs [106].

CTCs are a heterogeneous population of cells containing genomic, transcriptomic and proteomic features that reflect a patient’s tumor biology and can represent micrometastatic disease in vivo [126]. Mouse models with inoculated CTCs, i.e., CDX, provide an understanding of the mechanisms of cancer metastasis [127]. The response of CDX models to standard chemotherapy is consistent with the patient’s response rate, indicating their applicability for the development of novel therapeutics [128]. Serial CDX models established at different time points along with patient’s disease progression can be used for studying the mechanisms of the evolution of drug resistance [128]. Transcriptome sequencing of small cell lung cancer (SCLC) CDX revealed that the MYC signature negatively correlates with sensitivity to etoposide and platinum-based chemotherapy [128,129]. Single-cell sequencing of treatment-naïve and cisplatin-resistant SCLC CDX models found increased intra-tumoral heterogeneity, including the heterogeneous expression of therapeutic targets, between different cellular subpopulations following treatment resistance. In addition, the role of MYCL and NFIB genes in SCLC dissemination and the role of MYC and MYCL genes in drug resistance has been shown [129].

CTCs are injected at different sites depending on the type of cancer being studied, but most commonly in the heart, tail vein, or subcutaneously in the back for easy follow-up. CDX models have been established in breast, pancreatic, lung, and prostate cancers, and liver metastasis [82,113,128,130,131]. However, a sufficient number of viable CTCs is needed to develop in vivo models [132]. In this case, satisfactory in vivo models have been obtained with the RosetteSep isolation of CTCs [132]. In particular, melanoma CDX obtained from RosetteSep-isolated CTCs was used for studying patient response to the trametinib inhibitor of mitogen-activated protein kinases, MEK1 and MEK2 [133]. The issue of a small amount of CTCs is also overcome by using several passages from one mouse model to another [130]. For example, flow-sorted lineage-negative (CD45^−^CD34^−^CD105^−^CD90^−^CD73^−^) CTCs derived from peripheral blood samples from triple-negative breast cancer patients were injected intracardiacally in anesthetized NOD scid gamma mice. The obtained liver metastasis cells were injected intracardiacally into other mice [130]. Thus, this CDX model reproduces liver metastasis in serial CDX generations, displaying similar genomic profiles and the phenotypes of disseminated tumor cells [130].

Obtaining CTCs using the patient-derived xenografts (PDX) is another approach that compensates for the deficiency of methods for isolating CTCs directly from patients [129]. PDX models were originally generated from the primary tumors of patients. This approach has been successfully demonstrated using the tumors of patients with non-metastatic non-small cell lung cancer and castration-resistant prostate cancer [129,131]. CTCs obtained from PDX models are used to create CDX models, which are suitable for the investigation of drug response and metastasis as well as tumor cell heterogeneity. Such a method is useful for the development of a personalized strategy to combat metastases. It is known that the efficiency of creating PDX models is greater than that of CDX [126]. Therefore, obtaining a CDX model from PDX is a good methodological technique. However, PDX and CDX models only partially represent the patient’s tumor, as mice are characterized by a suppressed immune system that limits the study of metastasis processes [129].

In general, in vivo, ex vivo and in vitro models open up new opportunities in the field of translational and fundamental research (Figure 3). However, it should be noted that not all cells survive during the process of obtaining cell lines and xenografts, and some cells change genetically and phenotypically. Although CTC cell lines may share features with the primary tumor, there is a high probability that distinct clones will be selected [125]. High cell death in the bloodstream and a loss of some cells during isolation also limit the reliability and accuracy of CTC in vitro and in vivo models. Moreover, even successful isolation of viable CTCs may lead to obtaining non-target cells and to skewed results, especially in the case of different molecular analyses. For instance, our recent work showed a large number of non-target cells in samples enriched by the RosetteSep method [134]. Therefore, there is an urgent need to improve both CTC isolation methods and approaches for the development CTC in vivo, ex vivo and in vitro models.

## 5. Challenges and Trends

Progress in the isolation of CTCs has already been achieved in regard to technological developments and clinical oncology. All the methods described above can isolate viable CTCs for further research; nevertheless, they require improvement. Methods based on physical factors, such as the selection of cells by size, pass other tumor cells and clog filters, resulting in decreased throughput. Methods based on positive selection miss cells with different cellular phenotypes, e.g., CTCs in individual functional states (EMT-, mesenchymal- or stem-like). These methods also use a limited panel of markers that cannot cover the entire variety of CTCs. Methods based on negative selection also need to be refined due to the high content of non-target cells and because they skip hybrid CTCs, which are composed of two or more different cell types. Overall, all these methods should not affect CTCs by pH, pressure, and other physical factors, thereby isolating intact CTCs. It is important to note that due to the large phenotypic and genotypic diversity of CTCs, an ideal method for their isolation may never be developed.

Obviously, the genetic profile and functional phenotypes of CTCs, as well as their role in cancer metastasis and applicability as a model for screening and monitoring drug response, can only be completely revealed in the case of isolation of the maximum possible amount of CTCs. However, this issue still remains a great challenge because of the technical limitations discussed above. In addition, current methods are being developed to isolate CTCs from the venous circulation rather than arterial blood, which can be a better source of CTCs [135,136,137]. Another problem limiting the use of CTCs, especially in preclinical models, is high cell death during isolation. To achieve an increase in living target cells, technologies are needed to handle large sample volumes and improve the enrichment and purification by using both physical and biological CTCs parameters. Such an approach seems to be promising in regard to compensating for the disadvantages of different isolation methods, but it can be time-consuming and negatively influence the CTC viability.

Thus, an ideal CTC isolation method has not been established yet. Each of the available methods can be used for specific tasks and has its own advantages and disadvantages. For example, CTCs are rapidly isolated by the ieSCI-chip, an inertial microfluidic device, and spiral microfluidic technology. The simplest and cheapest methods are RosetteSep, NP@MNPs, and MNPs@hydrogel-anti-EpCAM. Other methods such as the CTC-iChip, Herringbone-Chip, and Neu-IMNs are highly efficient for obtaining CTC-derived cell lines. In general, the choice of method for CTC isolation should be based on the experimental objectives, accessible equipment, the researcher’s skills, funding, and other criteria.

## 6. Conclusions

There is still a crucial need to develop new technologies for the isolation of CTCs, especially living cells, which are of most interest for assessing anticancer drug efficacy and studying the mechanisms of cancer metastasis in vitro, ex vivo and in vivo models.

## Figures and Tables

**Figure 1 ijms-23-15979-f001:**
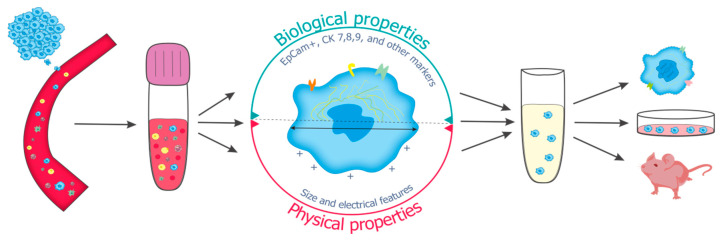
Cell properties used for viable CTC isolation and further applications. Viable CTCs are isolated from the blood sample using various methods based on biological, physical or combined features of cells and further used in molecular, in vitro, and in vivo research.

**Figure 2 ijms-23-15979-f002:**
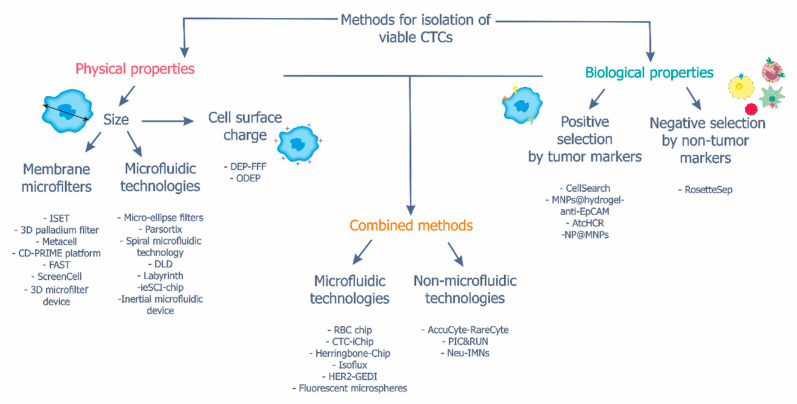
Methods for isolating viable CTCs. Viable CTCs can be isolated by methods based on physical (cell size and surface charge) and biological (specific markers) cell properties and combined methods that simultaneously consider the physical and biological features. The names of specific methods are provided.

**Figure 3 ijms-23-15979-f003:**
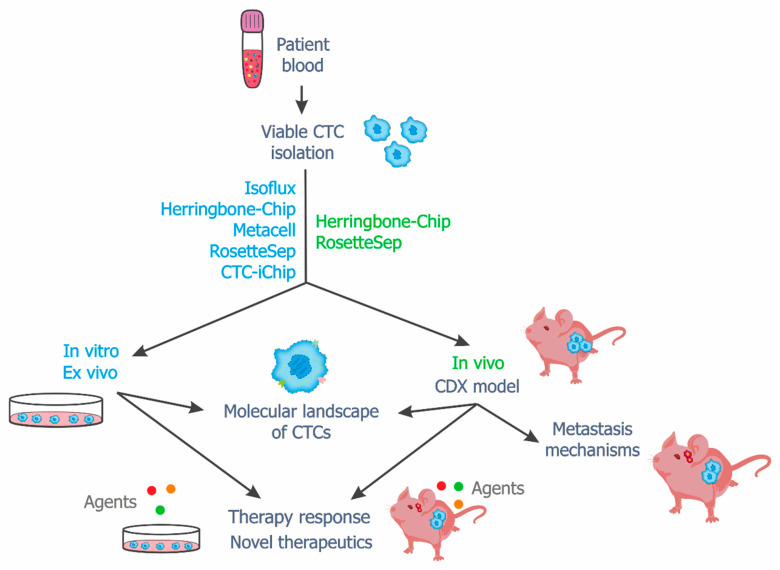
Potential applications of viable CTCs. The figure shows the methods that were efficient in obtaining CTC-derived in vitro, ex vivo and in vivo models. These models can be used for molecular analysis, drug response testing, the development of new therapeutics, and the investigation of metastasis mechanisms.

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
