# Peer review of "Technologies for Viable Circulating Tumor Cell Isolation"

_ijms, 2022, doi:10.3390/ijms232415979_

Round 1

Reviewer 1 Report (Previous Reviewer 3)

Dear Editor,

Thanks for your invitation again. After carefully reading through the resubmitted manuscript, I found that the new version has been well-revised. I have no more comments or advice to help promote the review work again. Furthermore, finally, I still suggest the Editor consider accepting the manuscript in its current form.

Author Response

We thank the Reviewer for the time spent reviewing our manuscript. Thanks for positive feedback.

Reviewer 2 Report (Previous Reviewer 2)

I disagree with the statement "The ScreenCell isolates pan-CK+CK7+CD45− cells" (pag. 5). It is clear that every cell, including leukocytes, having a diameter > 6-7 um does not pass through the isolation support. Furthermore, screencell devices isolate CTC not expressing CK as well.  Everything else is fine.

Author Response

We thank the Reviewer for the time spent reviewing our manuscript. Changes to the manuscript are marked by gray color.

We changed the sentence about ScreenCell:

“The composition of cells after isolation by the ScreenCell included cells with the pan-CK+/CK7+CD45 phenotype [68]. This method does not require expensive equipment and special operator skills.”

Reviewer 3 Report (Previous Reviewer 1)

In the manuscript "Technologies for viable circulating tumor cell isolation", the authors aim to summarize the knowledge on the current technologies  to isolate viable circulating tumor cells (CTC); however, all the methodologies described  isolate tumor cells regardless their viability. It is not emphasized over the most adequate technique to separate them or if the choice  will be determined by the tumor  type. It will be great if the reader could identify the best technique to extract viable CTCs (based on the tumor type) after reading this paper.

The point on "Particular attention to in vitro and in vivo models" is not justified because the objective o the review is to describe the available  techniques to isolate CTC with a high percentage of viability but no the utility of the CTC in future experiments. Figures are poorly described. 

Author Response

We thank the Reviewer for reviewing our manuscript and providing constructive criticism. Changes to the manuscript are marked by gray color.

The present review provides an overview of the current state of methods for isolating CTCs with high throughput and viability. Methods that are not suitable for obtaining viable CTCs, such as CellSearch, Ariol system, AdnaTest, and others, are not reviewed here. We emphasized this issue at the end of introduction.

A «Cancer type» column has been added to the tables.

We also expanded discussion about best methods for viable CTC isolation by adding the following information to section 5 «Challenges and trends»:

«Thus, an ideal CTC isolation method has not been established yet. Each of the available methods can be used for specific tasks and has its own advantages and disadvantages. For example, CTCs are rapidly isolated by ieSCI-chip, inertial microfluidic device, and spiral microfluidic technology. The simplest and cheapest methods are RosetteSep, NP@MNPs, and MNPs@hydrogel-anti-EpCAM. Other methods such as CTC-iChip, Herringbone-Chip, and Neu-IMNs are highly efficient for obtaining CTC-derived cell lines. In general, the choice of method for CTC isolation should be based on experimental objects, accessible equipment, researcher’s skills, funding, and other criteria».

We do not agree that the "Particular attention to in vitro and in vivo models" is not justified. This section demonstrates why CTC-derived models are important and how they can be used for understanding the molecular landscape of CTCs, testing therapy response, development of novel anticancer/antimetastatic drugs, and revealing metastasis mechanisms. A new figure (Fig. 3) demonstrates potential applications of viable CTCs and emphasizes again the importance of CTC-derived in vitro, ex vivo, and in vivo models.

The figure legends have been expanded.

This manuscript is a resubmission of an earlier submission. The following is a list of the peer review reports and author responses from that submission.

Round 1

Reviewer 1 Report

Tretiakova et al. summarize current techniques for isolating viable CTCs. I found that this work does not add to the current papers on CTCs isolation tools. Some technical limitations such as the total amount of blood sample needed for each method, collection conditions, time from sample collection to CTC extraction, etc. are not included. As well,  information on the best tools for CTC isolation in brain cancer is missing. Figures 1 and 2 do not provide relevant information and a table listing recent techniques, their applications, strengths and weaknesses would be more informative for readers.

Author Response

Reply: We thank the Reviewer for the time spent reviewing our manuscript. We appreciate your feedback and have revised the manuscript according to your suggestions.

We did not focus on the isolation methods of CTCs depending on cancer type, but, as suggested by the Reviewer, methods for isolating CTCs from glioblastoma patients (Müller Bark, J., et al. 2021, doi.org/10.3389/fonc.2021.681130; Zhang, H., et al. 2021 doi.org/10.3389/fonc.2021.607150) have been added in the revised manuscript. We also prepared three tables summarizing the methods for isolating CTCs, which show strengths and weaknesses, and further applications

Reviewer 2 Report

The aim of this review was to summarize the available literature regarding methods isolating viable circulating tumor cells (CTC), with particular attention to in vitro and in vivo models from CTC.

I have some comments to the manuscript:

- In the introduction section, authors should stress the utility of establishing in vitro and in vivo models from CTC, which is not limited to the development of therapy targeting metastasis.

- Several concepts are repeated throughout the text, as for instance, the biological and physical characteristics of CTC underlying isolation methods.

- The literature is replete with reviews regarding CTC insolation methods. So, it would be interesting to deepen advantages and disadvantages of such methods in order to obtain viable cells.

- Furthermore, authors should discuss section 4 in detail. Studies by Alix-Panabieres’ group are missing. Just to name one. There are not many who have succeeded in establishing CTC cultures after all.

I believe that a researcher approaching the CTC cultures should find useful information in this manuscript.

Author Response

Reply: We are very grateful that you took the time to review our work. We considered all comments and tried to answer your questions. New changes are marked by yellow. We combined figures 1 and 2, now there are 2 figures in the manuscript.

We didn't really mean just developing a therapy. Analysis of whole CTCs provides the ability to determine biological properties through in-depth in vitro and in vivo analysis, opening up new opportunities for basic and translational research. Cell lines and CDX from CTCs can lead to the identification of metastatic initiator cells. The corresponding text has been changed.

We didn't quite understand what you meant by «Several concepts are repeated throughout the text». This manuscript reviews CTC isolation methods based only on physical properties, only on biological properties, and physical-biological properties together. And in the text, one way or another, some text is repeated because some CTC methods are discussed several times in different sections of the manuscript from different points of view.

We prepared three tables summarizing the methods for isolating CTCs, which show strengths and weaknesses, and further applications.

We added works of Alix-Panabieres' group. The following text has been added as «Nine permanent lines of colon cancer were obtained from CTCs collected at different points in the treatment time using CellSearch. These cell lines showed different gene expression profiles and reflected changes in cell phenotypes during therapy».

Reviewer 3 Report

Circulating tumor cell isolation is a hot topic in recent years. In the authors' proposed manuscript, the technologies reported for circulating tumor cell isolation were generally well-addressed. While considering that similar contents and review frameworks have appeared several times in other journals, the reviewer suggests the authors choose a wholly new perspective to reconstruct the materials to attract audiences' attention. Besides that major comment, some minor comments were listed below.

1.      The advantage of "no-label" of the physical-based method was commonly stated as "label-free".

2.      Actually, the principle of CTCs isolation with DEP or oDEP methods mainly depends on the difference in their cell size. Meanwhile, the DEP and ODEP are similar techniques but only different in their approach to electrode charging. Thus, in figure 3, both the methods of DEP and oDEP need to be classified to "size" under "physical properties".

3.      Reviewing more new works reported in the recent five years will be better.

Author Response

Reply: First of all, we would like to thank the Reviewer for spending his/her effort and time reviewing our manuscript and providing constructive criticisms. We agree with the Reviewer that the addition of the important points raised will significantly improve our manuscript.

The main advantage of our manuscript is the focus on methods that allow isolating viable CTCs. In our opinion, such an approach is more informative and reliable due to the opportunity to perform single-cell multiomics, in vitro and in vivo studies.

We didn't understand the phrase "new perspective to reconstruct the materials". Could you please write more details what do you mean?

  1. We replaced "no-label" with "label-free".
  2. We took into account your comment and corrected Figure 3.
  3. We have added new methods and updated the references by adding several articles from the last 5 years.

Round 2

Reviewer 2 Report

I appreciate the efforts of the Authors to address my previous comments. However, some concerns to be addressed still remain

Major points:

- In the introduction section, authors should stress the utility of establishing in vitro and in vivo models from CTC. A single sentence is not enough.

- It would be interesting to deepen advantages and disadvantages of CTC isolation methods in order to obtain viable cells.

- Authors did not discuss section 4 in detail.

- Alix-Panabieres’ group enriched viable CTCs through depletion of hematopoietic cells using the RosetteSep Human Circulating Epithelial Tumor Cell Enrichment Cocktail and not using CellSearch System

Author Response

Reply:

- In the Introduction, we provided the detailed description to emphasize the importance of CTC in vitro and in vivo models as follow:

Obtaining viable CTCs enables creating preclinical models for drug screening and modeling of the metastatic cascade, including migration, invasion, and extravasation [7, 8]. CTC-derived cell lines open an opportunity to identify new markers for isolating CTCs from the blood and to develop additional predictive and prognostic criteria [9]. CTC cultures allow to evaluate the response to anticancer therapeutics and to individualize the treatment as well as open up doors in the development of drugs aimed at the metastasis prevention. CTC cell lines and xenografts (CDX) are also necessary to better understand the functional properties of CTCs obtained from cancer patients and to identify metastasis-initiating cells [10]. In general, CTC in vitro and in vivo models provide opportunities to determine biological properties of primary tumor and future metastatic cells opening new horizons for basic and translational research.

- We discussed section 4 in more detail:

In general, in vivo and in vitro models open up new possibilities in translational and fundamental research. However, it should be noted that not all cells survive during obtaining cell lines and xenografts, and some cells change genetically and phenotypically. Although CTC cell lines may share features with the primary tumor, there is a high probability of the selection of distinct clones [124]. High cell death in the bloodstream and a loss of some cells during isolation also limit reliability and accuracy of CTC in vitro and in vivo models. Moreover, even successful isolation of viable CTCs may lead to obtaining non-target cells and to skewed results especially in the case of different molecular analyses. For instance, our recent work showed a large number of non-target cells in samples enriched by the RosetteSep method [133]. Therefore, there is an urgent need to improve both CTC isolation methods and approaches for the development CTC in vivo and in vitro models.

- We deepen advantages and disadvantages of CTC isolation methods by adding new details to the text and in the tables.

- We corrected the sentence as suggested by the Reviewer:

Nine permanent lines of colon cancer were obtained from CTCs collected at different points in the treatment time using the RosetteSep Human Circulating Epithelial Tumor Cell Enrichment Cocktail.

Reviewer 3 Report

The reviewer recommanded that the current form of the manuscript was acceptable.